# HIF-1α Regulates Bone Homeostasis and Angiogenesis, Participating in the Occurrence of Bone Metabolic Diseases

**DOI:** 10.3390/cells11223552

**Published:** 2022-11-10

**Authors:** Wei Chen, Panfeng Wu, Fang Yu, Gaojie Luo, Liming Qing, Juyu Tang

**Affiliations:** Microsurgery & Reconstruction Research Center, Xiangya Hospital, Central South University, Changsha 410008, China

**Keywords:** hypoxia-inducible factor-1α, bone homeostasis, bone metabolic diseases, osteogenesis, angiogenesis, non-coding RNA, biomaterials, trace elements

## Abstract

In the physiological condition, the skeletal system’s bone resorption and formation are in dynamic balance, called bone homeostasis. However, bone homeostasis is destroyed under pathological conditions, leading to the occurrence of bone metabolism diseases. The expression of hypoxia-inducible factor-1α (HIF-1α) is regulated by oxygen concentration. It affects energy metabolism, which plays a vital role in preventing bone metabolic diseases. This review focuses on the HIF-1α pathway and describes in detail the possible mechanism of its involvement in the regulation of bone homeostasis and angiogenesis, as well as the current experimental studies on the use of HIF-1α in the prevention of bone metabolic diseases. HIF-1α/RANKL/Notch1 pathway bidirectionally regulates the differentiation of macrophages into osteoclasts under different conditions. In addition, HIF-1α is also regulated by many factors, including hypoxia, cofactor activity, non-coding RNA, trace elements, etc. As a pivotal pathway for coupling angiogenesis and osteogenesis, HIF-1α has been widely studied in bone metabolic diseases such as bone defect, osteoporosis, osteonecrosis of the femoral head, fracture, and nonunion. The wide application of biomaterials in bone metabolism also provides a reasonable basis for the experimental study of HIF-1α in preventing bone metabolic diseases.

## 1. Introduction

The skeletal system is an essential part of the human body. Under physiological conditions, osteoclasts and osteoblasts play a role in maintaining bone resorption and formation in dynamic balance [1]. Bone marrow mesenchymal stem cells (BMSCs) are important precursor cells of the osteoblast line [2]. Under appropriate conditions, BMSCs can differentiate into mesenchymal-derived cell types, such as chondrocytes, osteoblasts, and adipocytes, which play a crucial role in bone remodeling [3]. Osteoclasts derived from bone marrow-derived macrophages (BMMs) clear old or damaged bone under the trigger of osteocytes while releasing growth factors from the bone matrix, thus inducing BMSCs recruitment and migration to the repair area, resulting in BMSCs differentiation into osteoblasts and formation of new bone at specific sites [4,5]. However, under the pathological conditions of estrogen deficiency, abnormal mechanical stress, and drug side effects, the homeostasis of bone formation and bone resorption is disrupted, leading to osteoporosis, fracture, nonunion, or osteonecrosis of the femoral head [2]. Therefore, the regulation of bone formation and bone resorption is the basic strategy for preventing and treating bone metabolism-related diseases, in which osteoclasts and osteoblasts play a crucial role.

Hypoxia-inducible factor (HIF-1) is a pivotal transcriptional regulator of cell response to hypoxia, which is composed of HIF-1α and HIF-1β subunits, interacting with the hypoxia-responsive element (HRE) and further regulating the expression of target genes [6]. Under the normoxic condition, the proline residue of HIF-1α protein is hydroxylated by proline hydroxylase (PHD) in the cytoplasm with the participation of cofactors oxygen (O_2_), ferrous iron (Fe^2+^), and α-ketoglutaric acid. Then, it binds to the von Hippel–Lindau protein (VHL) and is rapidly degraded by the ubiquitin-proteasome system (UPS) [7,8,9]. However, hypoxia inhibits the hydroxylation of HIF-1α and induces HIF-1α to enter the nucleus, bind to HIF-1β, and initiate the transcription of hypoxia-related genes, such as vascular endothelial growth factor (VEGF) [10,11,12,13]. The biological function of HIF-1α has been widely studied in orthopedics fields. It participates in the occurrence and development of osteosarcoma, osteoarthritis, osteoporosis, fracture, osteonecrosis, and other diseases by affecting the osteoclast differentiation of BMMs and the osteogenic differentiation of BMSCs [14,15,16,17,18,19].

## 2. The Relationship between HIF-1α and Osteoclasts

Ferroptosis is a new form of regulatory cell death driven by iron-dependent lipid peroxidation, which participates in osteoclast differentiation [20,21]. Under normoxic conditions, ferroptosis could be induced due to the iron-starvation response (increased transferrin receptor 1, decreased ferritin) followed by the Receptor Activator of Nuclear Factor-κB Ligand (RANKL) stimulation [22]. Under hypoxia, HIF-1α upregulation inhibits ferroptosis by preventing BMMs autophagosome formation and promotes the expression of RANKL in osteoblasts by activating Janus kinase 2 (JAK2)/Signal Transducers and Activators of Transcription 3 (STAT3) pathway, which supports BMMs differentiating into osteoclasts, leading to osteoporosis [16,21,23,24]. However, Hulley et al. [25] believe that HIF-1α mainly acts as a regulator of osteoclast-mediated bone resorption and has almost no effect on osteoclast differentiation. In the inflammatory environment of high-energy metabolism, lysine-specific demethylase 1 (LSD1) induced by RANKL in a rapamycin-dependent manner stabilizes the expression of HIF-1α, thus promoting glycolysis and leads to pathological bone resorption [26]. Such an interaction between HIF-1α and RANKL plays a vital role in osteoclast differentiation and physiological/pathological bone resorption mediated by osteoclast.

Studies have shown that HIF-1α promotes osteocyte apoptosis by activating the c-Jun N-terminal kinase (JNK)/caspase-3 pathway, stimulates surrounding surviving osteocytes to synthesize cytokines such as RANKL and VEGF, and RANKL triggers Notch1 signal pathway necessary for BMMs to differentiate into osteoclasts, thus recruiting osteoclasts to clear dead cells and initiating remodeling of the surrounding matrix [27,28,29,30,31]. On the contrary, osteoprotegerin (OPG), a RANKL secreted non-signaling decoy receptor secreted by osteoblasts, can also be upregulated by HIF-1α and inhibit osteoclast-mediated physiological and pathological bone resorption [32,33]. In addition, the activation of the HIF-1α pathway in osteoblasts also promotes interleukin 33 (IL-33) expression. Then, IL-33 upregulates miR-34a-5p in a dose-dependent manner, followed by downregulation of osteoclast formation-related genes via the RANKL/Notch1 signal pathway, such as tartrate-resistant acid phosphatase (Trap), capthekin-K, nuclear factor of activated T cells c1 (NFATc1), and C-fos [34]. In addition, HIF-1α stimulates osteoclasts secreting Cardiotrophin-1 (CT-1), a gp130-signaling cytokine of the IL-6 superfamily, inducing BMSCs differentiating into osteoblast [35,36,37,38]. Chen et al. [39] stably expressed HIF-1α by knocking out the VHL gene in osteocytes and found that it could boost the osteogenic differentiation of BMSCs and damage the osteoclast differentiation of BMMs.

These results suggest that HIF-1α plays a dual role in the regulation of osteoclasts, which may be attributed to the maturity of osteoblasts (Figure 1). Mature osteoblasts reduce the destructive effect of osteoclasts around the sites where the bone is being removed by secreting OPG. In contrast, more immature osteoblasts promote bone resorption by expressing RANKL and a small amount of OPG [32].

## 3. Hypoxia Affects the Expression of HIF-1α and Regulates the Osteogenic Activity of BMSCs

### 3.1. Hypoxia Dual-Directionally Regulates the Expression of HIF-1α, Affecting the Proliferation and Osteogenic Differentiation of BMSCs

As oxygen concentration from the lungs to organs and tissues gradually decreases, the oxygen content around BMSCs becomes very low [40,41,42]. Related studies discussed the effects of oxygen concentration and hypoxia duration on the characteristics of human BMSCs. The results showed that the senescence of BMSCs cultured under the hypoxia condition of 5% O_2_ was inhibited, and the efficiency of osteogenic differentiation was improved [43,44,45,46,47]. In addition, HIF-1α enhanced the proliferation and stemness of peripheral blood-derived mesenchymal stromal cells (PBMSCs) under 5% O_2_ hypoxia conditions, accompanied by the multidirectional differentiation potential, including chondrogenesis, osteogenesis, and adipogenesis [48]. However, the expression of HIF-1α in BMSCs decreased under 1–2% O_2_ hypoxia conditions, activating the Notch1 and extracellular regulated protein kinase 1/2 (ERK1/2)/p38 mitogen-activated protein kinase (MAPK) signal pathways, which inhibited the osteogenic differentiation of BMSCs [49,50,51]. Furthermore, extreme hypoxia (0.2% O_2_) stimulates adipocyte differentiation rather than osteogenic differentiation of BMSCs by upregulating CCAAT enhancer-binding proteins (C/EBPs) and HIF-1α [52]. However, with the prolonged duration of hypoxia, the expression of HIF-1α decreases gradually, accompanied by core binding factor α1 (Cbfα1) increases, promoting human BMSCs differentiating into chondrocytes and osteoclasts [53,54,55]. In addition, hypoxia also activates HIF-1α through epigenetic mechanisms such as histone modification, DNA methylation, and regulation of non-coding RNA-related genes [56].

TWIST, as a transcriptional inhibitor of runt-related transcription factor 2 (RUNX2), is one of the downstream targets of HIF-1α, inhibiting BMSCs osteogenic differentiation by downregulating bone morphogenetic protein 2 (BMP2) and RUNX2 [57,58,59,60,61,62]. Mechano growth factor (MGF) is a splicing variant of insulin-like growth factor 1 (IGF-1), which can be used for autocrine tissue repair. Studies have shown that it promotes the growth and osteogenic differentiation of MSCs through the Phosphoinositide 3-kinase (PI3K)/protein kinase B (Akt) pathway [63,64]. Furthermore, the MGF E peptide reversed the low expression of HIF-1α under severe hypoxia via MEK-ERK1/2 and PI3K-Akt signal pathways, promoting BMSCs proliferation and osteogenic differentiation [65]. These results suggest that hypoxia degree and duration affect HIF-1α expression, regulating the biological behavior of BMSCs (Figure 2). It is unclear whether TWIST, as a critical factor in connecting HIF-1α and osteogenic genes, results in the above contradictory results, but the mechanism of MGF E peptide reversing HIF-1α low expression induced by severe hypoxia has been elucidated.

### 3.2. HIF-1α Regulates Bone Homeostasis by Affecting Energy Metabolism

Hypoxia leads to the insufficient energy production of mitochondria and excessive reactive oxygen species (ROS), which damages mitochondria. The high expression of HIF-1α transforms energy metabolism from oxidative phosphorylation to glycolysis, reduces ROS production, and enhances mitochondria’s tolerance to hypoxia injury, suggesting that hypoxia plays a regulatory role in energy metabolism by inducing adaptive high expression of HIF-1α [66,67]. HIF-1α is associated with various diseases by affecting the glycolysis pathways of tumors, inflammation, and immune cells and regulating biological processes such as cell proliferation, differentiation, migration, chemotaxis, phagocytosis, and apoptosis.

Studies have shown that energy metabolic genes, including glucose transporter 1 (GLUT1), pyruvate dehydrogenase kinase 1 (PDK-1), lactate dehydrogenase (LDH), and monocarboxylate transporter-4 (MCT-4), upregulate in a hypoxia environment [68,69]. HIF-1α promotes glycolysis in the bone marrow microenvironment by upregulating the key enzyme PDK1 and then induces osteoblast differentiation, stimulating bone formation [70,71]. The enhancement of glycolysis in VHL-deficient osteoblasts increases bone mass, suggesting that HIF-1α affects bone homeostasis by regulating energy metabolism [72]. In addition, HIF-1α up-regulates in the early stage of BMSCs differentiation, promoting glycolysis to provide energy for BMSCs proliferation, but significantly downregulates in the late stage of BMSCs differentiation, enhancing oxidative phosphorylation in both osteogenic and chondrogenic-induced BMSCs [73,74,75]. These results somewhat explain the opposite effects of HIF-1α in bone energy metabolism in different studies. Some studies have pointed out that HIF-1α inhibits load-induced bone formation by activating the AKT/mammalian target of rapamycin (mTOR) pathway and inhibiting the Wnt/β-catenin pathway, changing osteoblast and osteocyte sensitivity to mechanical signals, which plays a negative regulatory role in the anabolism induced by bone mechanical load [71,76,77,78].

## 4. Inactivation of PHD/VHL Stabilizes HIF-1α, Promoting Angiogenesis and BMSCs Osteogenic Differentiation

### 4.1. PHD

Cobalt ions (Co^2+^) inhibit HIF-1α degradation by inactivating PHD and preventing VHL proline residues from being bound to HIF-1α [79,80]. Therefore, Co^2+^ stabilizes HIF-1α expression and upregulates VEGF by inducing ROS production and activating HRE [81]. Co^2+^ stimulates STAT3 phosphorylation and HIF-1α expression in BMSCs, inducing BMSCs migration to the bone repair zone [82,83]. Moreover, Co^2+^ also downregulates proliferation genes in BMSCs, including cyclin D1, while upregulating HIF-1α target genes, including erythropoietin (EPO), VEGF, and p21 [84]. Based on these studies, Co^2+^, as a chemical inducer of HIF-1α, plays a regulatory role in BMSCs migration and proliferation. Furthermore, Co^2+^ promotes osteogenesis and angiogenesis by stabilizing HIF-1α, which supports its application and development in the basic research of bone metabolism-related diseases [83,85,86,87].

Deferoxamine (DFO), a chelating agent, has been widely used in the clinical treatment of iron overload diseases [88,89,90]. Like Co^2+^, DFO chelates with non-protein bound iron and inhibits Fe^2+^-dependent PHD activity to simulate hypoxia, thus inhibiting the hydroxylation and degradation of HIF-1α [91,92,93]. DFO reduces ROS production in mitochondria and cell apoptosis in a HIF-1α dependent manner and promotes tubular formation, proliferation, and migration of endothelial cells (ECs) by promoting VEGF expression [94,95,96,97,98,99,100]. Short-term DFO treatment significantly changes the BMSCs glycolysis pathway, reduces mitochondrial oxidative stress, and upregulates nuclear protein1 (NUPR1) in a concentration-dependent manner, promoting BMSCs survival and cell protective autophagy [91]. In addition, DFO also affects histone acetylation and DNA methylation-related genes in osteoblasts through epigenetic mechanisms [101]. Previous studies have shown that DFO reduced the adverse effects of radiation on angiogenesis and distraction osteogenesis [102,103]. To go a step further, DFO promotes mineralization, angiogenesis, and osteogenic differentiation of BMSCs by stabilizing and enhancing HIF-1α [104]. As a HIF-1α stabilizer, DFO has been widely studied in promoting osteoblast activity and inhibiting osteoclast differentiation. However, its clinical application in bone metabolism diseases needs to be further explored [105,106,107,108,109,110].

Ethyl 3,4-dihydroxybenzoate (EDHB) is a small molecule antioxidant, which inhibits PHD activity, stabilizes HIF-1α, and promotes angiogenesis by upregulating VEGF, thus preventing steroid-induced osteonecrosis of the femoral head [18,111]. As a PHD inhibitor, Roxadustat promotes bone formation and inhibits bone resorption by activating HIF-1α and Wnt/β-catenin signal pathways, improving bone microstructure deterioration and reducing bone loss in ovariectomized rats [16,112]. IOX2 and dimethyloxalylglycine (DMOG), as inhibitors of PHD, reduce the degradation of HIF-1α, regulate the expression of downstream target genes of HIF-1α, prompting angiogenesis and the proliferation, migration, and osteogenic differentiation of BMSCs, as well as inhibiting osteoblast apoptosis [17,113,114,115,116,117,118]. However, the long-term duration activation of HIF-1α by DMOG and baicalein inhibits the osteogenic differentiation and enhances angiogenesis of adipose-derived stem cells (ASCs), while the stem cell characteristics are preserved, which still shows superiority in repairing rat skull defects [119,120] (Figure 3).

### 4.2. VHL

VHL affects HIF-1α expression by binding and degrading the hydroxylated HIF-1α protein. Inactivated VHL gene stimulates osteochondral progenitor cells to differentiate to osteoblasts via upregulating HIF-1α/VEGF and Smad1/5/8 signal pathways in newborn mice, accompanied by increased differentiation markers Runx2 and osteocalcin (OCN), showing significant cancellous bone accumulation, microvascular density, and bone formation increase [121]. BMP9, expressed in the developing mouse liver, induces MSCs to express HIF-1α through the Smad1/5/8 signal pathway and plays a synergistic role with HIF-1α in the process of inducing osteogenic differentiation and angiogenesis [122,123,124,125,126]. These results suggest that VHL inactivation may upregulate HIF-1α by inducing the Smad1/5/8 signal pathway, which may be a new mechanism by which VHL participates in HIF-1α regulation (Figure 3).

Selective knockout of the VHL gene up-regulates HIF-1α in mouse osteoblasts. It promotes bone marrow angiogenesis, showing a significant increase in trabecular volume, which effectively reverses bone loss caused by estrogen deficiency and promotes long bone formation with high bone density and rich blood supply [127,128,129]. Zuo et al. [130] found that the cortical bone area of VHL-deficient mice increased significantly, which was attributed to the proliferation and osteogenic differentiation of BMSCs promoted by the VHL/HIF-1α/β-catenin pathway. In addition, VHL deficiency of osteocytes increases bone mass and bone marrow hematopoiesis by regulating HIF-1α/Wnt signal pathway [131]. These studies suggest that VHL/HIF-1α pathway is a pivotal mediator in regulating angiogenesis and osteogenic differentiation of BMSCs. In addition, VHL also plays a vital role in regulating bone morphogenesis. Although the absence of VHL does not change the differentiation direction of mesenchymal progenitor cells into chondrocytes during development, it damages the proliferation ability of chondrocytes, resulting in the structural collapse of the growth plate and affecting bone morphogenesis [132,133].

## 5. HIF-1α/VEGF Pathway Regulates Type H Vessels, Coupling of Angiogenesis and Osteogenesis

### 5.1. HIF-1α/VEGF Pathway Connects Type H ECs and Osteoblasts in the Skeletal System

As the target gene of HIF-1α, VEGF plays an essential role in the regulation of angiogenesis [10,11,12,13,134,135]. Studies have shown that HIF-1α/VEGF pathway is involved in tumor immunity, inflammation, ischemia-reperfusion injury, oxidative stress, and other angiogenesis or vascular remodeling pathophysiological processes [136,137,138,139,140,141,142,143]. The mechanism of the HIF-1α/VEGF pathway involved in the regulation of bone homeostasis has also been widely studied, including angiogenesis–osteogenesis coupling. Under the mutual regulation of angiopoietin-1 (Ang-1) and angiopoietin-2 (Ang-2), HIF-1α stimulates BMSCs to secrete VEGF and inhibits the expression of tissue inhibitor of metalloproteinase-3 (TIMP-3), an endogenous competitive inhibitor of VEGF receptor, which mediates osteogenesis and angiogenesis. This mutual regulation relationship is known as the “HIF-1α-VEGF-Ang-1 axis” [144,145,146,147,148].

Type H ECs are a subtype of vascular endothelial cell with high expression of CD31 and Endomucin (EMCN) in the metaphyseal, distributing many bone progenitor cells around, which can differentiate into osteoblasts and osteocytes [149,150]. Two consecutive studies in *Nature* by the same team in 2014 found that Type H ECs mediate local vascular growth via HIF-1α/VEGF and provide clear signals for perivascular osteoblasts through the Notch signal pathway [150,151]. HIF-1α is highly expressed in Type H ECs of young rats, decreasing with aging, and is related to age-dependent bone loss, which can be reversed by DFO [152]. EC-specific inactivated VHL gene enhances Type H vessels angiogenesis and increases the number of bone progenitor cells by stabilizing HIF-1α in ECs [8]. In addition, hypoxia signals increase the number of Type H vessels and enhance endochondral angiogenesis and osteogenesis [150]. Osteostatin (OST), mainly expressed in bone marrow, induces BMSCs osteogenic differentiation and promotes the proliferation, migration, and angiogenesis of Type H ECs via the HIF-1α/VEGF pathway under hypoxia conditions [153,154]. These results suggest that HIF-1α/VEGF pathway regulates angiogenesis and osteogenesis by connecting Type H ECs and osteoblasts in the skeletal system (Figure 4).

### 5.2. HIF-1α/VEGF Pathway Regulates Type H Vessels in Various Bone Metabolic Disease Models

Studies have shown that the number of Type H vessels in the long bone of aging mice decreased significantly, accompanied by bone progenitor cells decrease. The osteoporosis model of ovariectomized mice showed the same phenotype [152]. Local injection of tetramethylpyrazine into the bone marrow of aging mice directly induces the formation of Type H vessels. It improves bone homeostasis by regulating the AMPK/mTOR and HIF-1α/VEGF signal pathways, preventing and treating glucocorticoid-induced osteoporosis [155]. In addition, miR-210 significantly promotes HIF-1α and VEGF expression in BMSCs dose-and-time-dependently, upregulates alkaline phosphatase (ALP) and Osterix, inhibits peroxisome proliferators-activated receptors (PPARγ), and induces BMSCs differentiating into osteoblasts, improving osteoporosis caused by estrogen deficiency [156]. Similarly, miR-497~195 cluster improves senile osteoporosis by inducing Type H vessel angiogenesis coupling with osteogenesis via Notch and HIF-1α pathways [157]. In addition, matrix metalloproteinase-2 inhibitor 1 (MMP2-I1) could induce BMSCs osteogenesis differentiation and promote Type H vessel angiogenesis through the HIF-1α signal pathway, rescuing osteonecrosis, bone defect, as well as osteoporosis [158,159] (Figure 4).

**Figure 4 cells-11-03552-f004:**
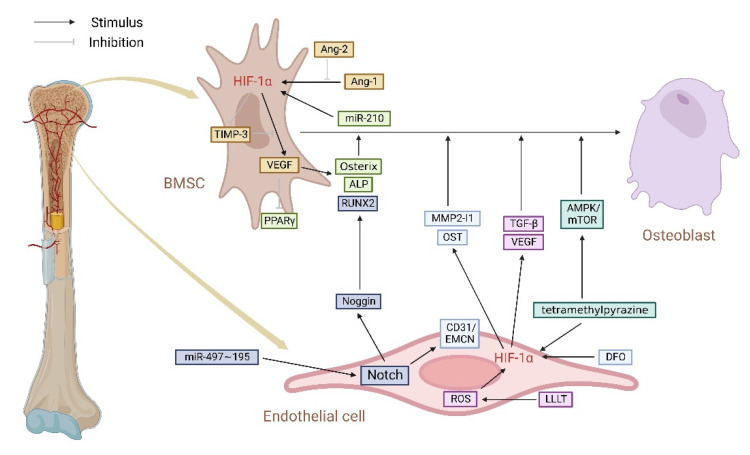
HIF-1α couples angiogenesis and osteogenesis. HIF-1α/VEGF and Notch signal pathways are pivotal in coupling Type H vessels angiogenesis and BMSCs osteogenic differentiation, in which matrix metalloproteinase-2 inhibitor 1 (MMP2-I1) and Osteostatin (OST) play a positive regulatory role. In addition, the HIF-1α/VEGF pathway also promotes alkaline phosphatase (ALP) and Osterix, inhibits peroxisome proliferators-activated receptors (PPARγ), promotes BMSCs osteogenic differentiation, and participates in the occurrence of osteoporosis, osteonecrosis, and bone defect.

Low-level laser therapy (LLLT) has a positive photobiological stimulation effect on cell proliferation, angiogenesis, osteogenic differentiation, bone regeneration, and fracture healing [160,161,162]. The relevant evidence shows that laser irradiation changes the mitochondrial membrane potential, promotes the oxidation of ferrous iron by inducing ROS production, and then inhibits HIF-1α degradation by inactivating the PHD, accompanied by VEGF and transforming growth factor-β (TGF-β) expression upregulation, promoting Type H vessel formation and BMSCs osteogenic differentiation, coupling angiogenesis and osteogenesis [163,164,165] (Figure 4).

## 6. The Role of Non-Coding RNA (ncRNA) in HIF-1α Regulating Bone Homeostasis

### 6.1. Interaction between miRNA and HIF-1α Promotes Angiogenesis and BMSCs Osteogenic Differentiation, Preventing Bone Metabolism Diseases

MicroRNA (miRNA) is a group of single-stranded endogenous ncRNA with a length of 20–24 nucleotides. It plays a role in maintaining bone homeostasis by regulating angiogenesis and the BMSCs differentiation direction and may be used as a diagnostic marker for bone metabolism diseases [166]. Studies have shown that miRNA-21 promotes BMSCs expressing the osteogenic protein dose-dependently, including BMP-2, Runx2, OCN, and OPN, which may be related to phosphate and tension homology deleted on chromosome ten (PTEN)/PI3K/Akt/HIF-1α pathway [167]. MiRNA-675-5p upregulates Wnt/β-catenin and HIF-1α/VEGF signal pathways under hypoxia conditions, stimulating MSCs stemness markers (CD44, CD90, and CD73) downregulation and early osteoblast markers (ALP and COL1A1) upregulation, and then induces MSCs osteogenic and chondrogenic differentiation and promotes angiogenesis [15,168]. Low-intensity pulsed ultrasound (LIPUS) upregulates miR-31-5p, then promotes HIF-1α expression and Ras Homolog Family Member A (RhoA) activation through both VHL-dependent and VHL-independent pathways, upregulating RUNX-2, ALP, OCN, and secreted phosphoprotein 1 (SPP1), inducing human MSCs migration and promoting angiogenesis and bone regeneration [169]. Under hypoxia conditions, miR-210-3p in MSCs-derived extracellular vesicles (EVs) was regulated by HIF-1α, which blocked tyrosine kinase ligand ephrin-A3 (EFNA3) expression, then significantly promoted the angiogenesis by activating the PI3K/AKT pathway [23]. In addition, miR-26a-5p in EVs derived from urine-derived stem cells (USCs) activates HIF-1α/VEGF pathway by inhibiting histone deacetylase 4 (HDAC4), then promotes the differentiation of osteoblast progenitor cells and inhibits the activity of osteoclasts, thus preventing the occurrence of diabetic osteoporosis [170] (Figure 3).

### 6.2. Interaction between lncRNA and HIF-1α Regulates Apoptosis, Angiogenesis, and BMSCs Osteogenic Differentiation Involved in Bone Development and Regeneration

Long non-coding RNA (lncRNA) plays a vital role in cell proliferation, invasion, metabolism, apoptosis, and stem cell differentiation. LncRNA participates in bone development and regeneration by regulating apoptosis, angiogenesis, and BMSCs differentiation [171]. The gene sequencing results showed that 95 lncRNAs were differentially expressed in osteoclasts knocked out of HIF-1α, suggesting that lncRNA may play a role in bone homeostasis regulation by HIF-1α [172]. Metastasis-associated lung adenocarcinoma transcript 1 (MALAT1), as a highly conserved lncRNA, induces angiogenesis through the mTOR/HIF-1α pathway [173]. Enhancer of zeste homolog 2 (EZH2) is a catalytic subunit of histone methyltransferase, which catalyzes the histone H3 lysine 27 trimethylation (H3K27me3) [174]. Under hypoxia conditions, HIF-1α binds to the HRE of the EZH2 promoter, then suppresses the expression of LncRNA-Tmem235 and downregulates miR-34a-3p/baculoviral inhibitor of apoptosis repeat-containing 5 (BIRC5), resulting in anoxic apoptosis of BMSCs [175]. In addition, TGF-β in BMSCs induces deacetylase sirtuin 1 (SIRT1) overexpression, downregulates lncRNA-HIF-1α-AS1 and blocks homeobox (HOX) D10 expression to interfere with acetylation, thus inhibiting osteoblast differentiation [176] (Figure 3).

## 7. HIF-1α Involvement in the Bone Repair of Biomaterials

### 7.1. Biomaterials Doped with Trace Elements Upregulate HIF-1α, Promoting Angiogenesis and Osteogenesis

#### 7.1.1. Copper

Biomaterials doped with trace elements as scaffolds have attracted much attention in bone regeneration. The most common trace elements are copper (Cu), lithium (Li), cobalt (Co), zinc (Zn), magnesium (Mg), titanium (Ti), silicon (Si), etc. Studies have shown that continuous release of Cu^2+^ and Li^2+^ from organic or inorganic scaffolds causes crosstalk between Wnt and HIF-1α signal pathways, coupling angiogenesis and BMSCs osteogenic differentiation [177]. Hydroxyapatite (HA), as a commonly used inorganic biomaterial, loaded with co-overexpressed osteogenic factor semaphoring 3A (Sema3A) and HIF-1α modified MSCs, promotes osteogenic differentiation [178,179]. Furthermore, Nano-HA stimulates ECs to produce HIF-1α dose-dependently and upregulate osteogenic genes via the ERK1/2 signal pathway, increasing mineralized nodules and ALP [180]. Cu-Li-doped Nano-HA (Cu-Li-nHA) upregulates HIF-1α and stromal cell-derived factor-1 (SDF-1) by steadily releasing Cu^2+^, then promotes angiogenesis and induces BMSCs homing to necrotic areas, stimulating BMSCs to differentiate into osteoblasts through Wnt/GSK-3β/β-catenin pathway [181,182,183].

Titanium dioxide (TiO_2_) coating containing Cu not only improves biomedical implants’ antibacterial ability but also promotes HIF-1α/VEGF and osteogenic markers expression in BMSCs, thus achieving antibacterial properties, osteogenesis, and angiogenesis [184]. Water-soluble graphene oxide-Cu nanocomposite (GO-Cu) stably releases Cu^2+^, activating the ERK1/2 signal pathway and upregulating HIF-1α/VEGF and BMP-2 in BMSCs [185]. Furthermore, porous calcium phosphate (CaP) scaffold coated on GO-Cu significantly promoted angiogenesis and osteogenesis, repairing the bone defect of the rat skull. Collagen-coated Cu-doped calcium polyphosphate (CCPP) scaffolds stably release Cu^2+^ and collagen coating, induce BMSCs proliferation and migration to the bone defect area, then upregulate HIF-1α/VEGF and osteogenesis-related genes, stimulating BMSCs osteogenic differentiation [186].

PH-neutral bioactive glass (PSC) induces angiogenesis and BMSCs osteogenic differentiation through the PI3K/AKT/HIF-1α pathway. Compared with traditional 45S5 bioactive glass (BG) and β-tricalcium phosphate (β-TCP), PSC has more advantages in promoting BMSCs proliferation, migration, mineralization, and angiogenesis [187]. Cu-containing BG (Cu-BG) not only supports BMSCs attachment and expansion but also continuously activates HIF-1α and tumor necrosis factor-α (TNF-α) by stably releasing Cu^2+^, promoting downstream vascular-related growth factors and inflammation-related factors expression, directly inducing BMSCs differentiating into osteoblasts [188,189].

#### 7.1.2. Cobalt

Co^2+^ stabilizes HIF-1α by inactivating PHD in a normoxic environment, commonly used as a hypoxia inducer [80]. It has been confirmed that BMSCs treated with Co^2+^ promote osteogenesis and angiogenesis in bone defect areas [85]. Co-doped HA significantly upregulates HIF-1α and VEGF in MG-63 cells, then promotes cell cycle progression and proliferation and induces osteocyte differentiation [190]. HA as a prosthesis coating triggers survival osteogenic gene signals, rescues the inhibition of osteoblasts and osteoclasts activity caused by Co^2+^ and chromium (Cr) ions concentration elevation around the prosthesis through regulating the HIF-1α signal pathway and endocytic/cytoskeletal gene [191]. Co-doped BG scaffolds with different mesoporous or chemical modifications of the upregulation of VEGF, HIF-1α, and osteogenic genes in BMSCs, promote angiogenesis and osteogenic differentiation, as well as inhibit chondrogenic differentiation [86,192,193]. The hydrogel fiber scaffold composed of collagen and alginate achieves a more substantial bone repair by carrying Co^2+^ and BMP2 [194]. In addition, combined organic and inorganic biomaterials doped with Co^2+^, such as collagen glycosaminoglycan (CG)-BG and calcium alginate (CA)-Nano-HA, also have an excellent ability to promote bone tissue regeneration [195,196].

#### 7.1.3. Other Trace Elements

Zn commonly presents in all tissues, body fluids, and organs of the human body—more than 95% of which are present in cells and involved in the transcription of DNA [197]. Under 1% O_2_ conditions, Zn^2+^ inhibits HIF-1α expression, promoting migration and proliferation and delaying the senescence of BMSCs [198]. Ti scaffolds coated on Mg significantly stimulate the proliferation, adhesion, extracellular matrix mineralization, and ALP activity of MC3T3-E1 cells via the JAK1/STAT1/HIF-1α pathway, as well as improve the proliferation, adhesion, tubular formation, scratch healing and Transwell ability of human umbilical vein endothelial cells (HUVECs) through HIF-1α/VEGF pathway [199,200]. Zn/Mg co-implanted Ti scaffold (Zn/Mg-PIII) upregulates magnesium transporter 1 (MAGT1) in HUVECs, promotes Mg^2+^ influx, and activates the HIF-1α/VEGF pathway, inducing angiogenesis. In addition, Zn/Mg-PIII upregulates integrin α1 and integrin β1 to promote rat BMSCs adhesion and spread, promoting Runx2, ALP, and OCN expression in BMSCs by inducing Zn^2+^ and Mg^2+^ to recruit into cells. Compared with traditional Ti scaffolds, the Zn/Mg-PIII enhanced bone formation, angiogenesis, and antibacterial activity [201]. The ions released from Si-based biomaterials upregulate HIF-1α. The porous TiO_2_ coating doped with a small amount of Si significantly enhances HUVECs angiogenesis, while excessive Si doping will impair the vascular response [202]. Monte et al. [203] confirmed that 0.5mM, as an appropriate concentration of Si^4+^, enhanced HUVECs viability by alleviating oxidative stress damage and promoted HUVECs proliferation, migration, and angiogenesis through upregulating HIF-1α. Furthermore, mesoporous silica nanoparticles (MSN) loaded with platelet-derived growth factor BB (PDGF-BB) significantly stimulate the “HIF-1α-VEGF-Ang-1 axis” and up-regulate osteogenesis-related genes in BMSCs, inducing osteogenesis and angiogenesis [204].

These results suggest that suitable biomaterials induce BMSCs migration and attachment to the bone repair zone by stably releasing trace elements such as Cu^2+^ and Co^2+^ and upregulate HIF-1α/VEGF and osteogenic genes, coupling angiogenesis and osteogenesis, achieving bone regeneration (Figure 5).

### 7.2. Biomaterials Loaded with PHD Inhibitors Stabilize HIF-1α, Promoting Bone Defect Repair

#### 7.2.1. DFO

As an iron chelating agent, DFO stabilizes HIF-1α by inhibiting PHD activity. Poly (lactic-co-glycolic acid) (PLGA) loaded with DFO significantly stimulates angiogenesis and BMSCs osteogenic differentiation, promoting osteoporotic bone defect repair [205]. DFO also effectively induces bone regeneration in true bone ceramic (TBC) scaffolds, promoting angiogenesis and segmental bone defect repair [206,207]. Titania nanotube (TNT) diameter dependently enhances BMSCs proliferation and mineralization ability. As a drug carrier loaded with DFO, TNT promotes angiogenesis and BMSCs osteogenic gene expression by activating the HIF-1α signal pathway by continuously and stably releasing DFO, then stimulates BMSCs adhesion and proliferation and affecting HUVECs growth behavior [61,208]. DFO also significantly improves the surface roughness and hydrophilicity of the polydopamine (PDOPA) membrane, which is more conducive to the attachment, proliferation, and spread of MC3T3-E1 cells and HUVECs, thus achieving bone repair [209] (Figure 5).

#### 7.2.2. DMOG

DMOG, as an inhibitor of PHD, reduces HIF-1α degradation. β-TCP loaded with DMOG significantly enhances the angiogenic activity of BMSCs, thus repairing skull defects [210]. Implantation of ASCs on porous β-TCP-alginate-gelatin scaffolds containing DMOG induces angiogenesis and bone repair in rat skulls [211]. MSN slowly releases Si^4+^ to activate ALP, OCN, RUNX2, and OPN expression in human BMSCs and to stimulate BMSCs osteogenic differentiation, while MSN loaded with DMOG upregulates angiogenic genes in BMSCs by stabilizing HIF-1α, coupling osteogenesis and angiogenesis, which play a role in repairing bone defects [212]. Similarly, mesoporous bioactive glass (MBG) doped with DMOG significantly promotes osteogenesis and angiogenesis [213,214,215]. In addition, different mesoporous MBGs loaded with DMOG stabilize HIF-1α continuously by controlling DMOG release, promoting VEGF and osteogenic gene expression in BMSCs, which achieves more obvious angiogenesis and osteogenic effect [216] (Figure 5).

### 7.3. Application of Tissue Engineering Combined with Gene Mutation Technique to HIF-1α in Bone Defects Repair

Angiogenic and osteogenic gene upregulation in normoxic conditions can be achieved by transplanting HIF-1α-modified BMSCs into bone defect areas [217]. Studies have shown that gene mutation technology effectively improves HIF-1α activity, promoting the application in bone metabolic diseases. Compared with wild-type HIF-1α, the EVs derived from BMSCs modified by mutant HIF-1α achieve a better osteogenic and angiogenic ability, thus rescuing steroid-induced osteonecrosis of the femoral head [218]. Furthermore, β-TCP scaffold loaded the rat BMSCs-derived EVs carrying mutant HIF-1α promotes angiogenesis and BMSCs proliferation and osteogenic differentiation [219]. Ca-Mg phosphate cement (CMPC) scaffold carrying the structural active form (CA5) of HIF-1αachieves higher osteogenic activity in repairing bone defects [220,221]. The gelatin sponge (GS) loaded the BMSCs transfecting CA5 upregulates angiogenic genes in BMSCs, promoting angiogenesis in the bone defect area [222]. These results suggest that tissue engineering combined with gene mutation effectively promotes bone regeneration and may be used as an effective treatment for bone metabolism diseases.

## 8. The Role of Other Factors in Bone Homeostasis Regulation by HIF-1α

Toll-like receptor 2 (TLR2) is essential in regulating the immune response. Previous studies have shown that TLR2 activation promotes tissue angiogenesis and wound healing [223]. Furthermore, TLR2 significantly enhances HIF-1α and BMP-2 expression in BMSCs, upregulating the downstream osteogenic and angiogenic genes [224]. Epidermal growth factor (EGF) participates in proliferation, differentiation, adhesion, and survival processes. As a ligand of EGF, betacellulin (BTC) stabilizes HIF-1α, stimulating osteogenic progenitor cell proliferation while inhibiting differentiation [225,226]. As a demethylase, fat mass and obesity-associated protein (FTO) regulates the balance between adipogenic and osteogenic differentiation of BMSCs [227]. However, FTO induces BMSCs osteogenic differentiation by upregulating HIF-1α under mechanical stress conditions [228].

Icariin (ICA) is the main active component of Epimedium, which induces BMSCs migration by activating the SDF-1α/HIF-1α/CXCR4 pathway [229]. Safflower yellow (SY), as the main component of the traditional Chinese medicine safflower, promotes blood circulation and removes blood stasis [230]. In addition, SY also plays a pivotal role in osteogenesis and angiogenesis via the VHL/HIF-1α/VEGF signal pathway [231]. Curcumin, which exists in turmeric, significantly rescues hypoxia and reoxygenation (H/R) damage of BMSC through inhibiting mitochondrial ROS accumulation, which may be related to HIF-1α instability, the exchange protein activated by cAMP-1 (Epac1) and Akt activation and ERK1/2 and p38 inactivation [232]. Salidroside (SAL), as the main bioactive component of Rhodiola, significantly promotes the proliferation, migration, and angiogenesis of HUVECs through the HIF-1α/VEGF pathway and induces EC germination from the metatarsal [233]. Statins are traditional lipid-lowering drugs that promote fracture healing and bone defect repair [234,235,236]. Furthermore, simvastatin may induce BMSCs and endothelial progenitor cell migration via the HIF-1α/BMP-2 pathway, promoting bone defect healing [237].

## 9. Conclusions

Currently, the clinical treatment of bone metabolic diseases includes surgery and drug therapy. Bone transplantation in treating bone defects, joint prosthesis replacement in treating osteonecrosis of the femoral head, and plates and screws in treating fractures have achieved good results. However, it is difficult for drugs to reverse osteoporosis and prevent complications. Therefore, it is urgent to study the mechanism of bone metabolic disorder and implement effective intervention measures. With the effects of HIF-1α having been widely studied, its role in angiogenesis and bone homeostasis regulation has aroused the interest of researchers. HIF-1α regulates angiogenesis and osteoclasts and osteoblasts differentiation in various mechanisms, in which the epigenetic regulation of ncRNA also plays an important role. Based on the beneficial effects of HIF-1α coupling angiogenesis and osteogenesis and the application of biomaterials in bone metabolic diseases, new scaffold materials loaded with trace elements or drugs involved in the HIF-1α pathway may reach a better bone regeneration effect. Although HIF-1α has been widely studied in bone metabolism, there are still some problems to be explored. The bidirectional regulation of hypoxia on HIF-1α expression and the effect of HIF-1α on osteoclast differentiation have not been fully elucidated. It is necessary to study the mechanism of these opposite effects further and explore the effectiveness of HIF-1α in preventing bone metabolic diseases, laying a foundation for further clinical application.

## Figures and Tables

**Figure 1 cells-11-03552-f001:**
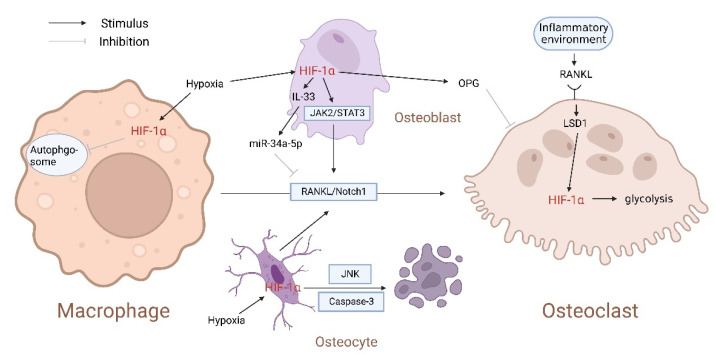
Hypoxia-inducible factor-1α (HIF-1α) plays a bi-directional regulatory role in differentiating macrophages into osteoclasts. Nuclear Factor-κB Ligand (RANKL)/Notch1 is a crucial signal pathway for inducing macrophages to differentiate into osteoclasts. Hypoxia upregulates HIF-1α, inhibiting autophagosomes formation in macrophages, promoting the expression of RANKL by activating the Janus kinase 2 (JAK2)/Signal Transducers and Activators of Transcription 3 (STAT3) pathway in osteoblast. In addition, HIF-1α also activates the c-Jun N-terminal kinase (JNK)/caspase-3 pathway in osteocytes and stimulates the RANKL/Notch1 signal pathway. However, miR-34a-5p inhibits RANKL/Notch1 signal pathway under HIF-1α/interleukin 33 (IL-33) stimulation.

**Figure 2 cells-11-03552-f002:**
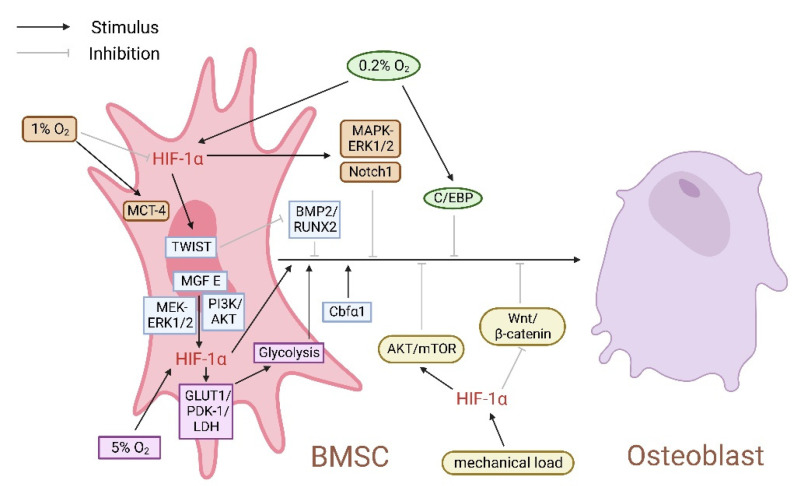
Hypoxia affects the differentiation of bone marrow mesenchymal stem cells (BMSCs) into osteoblasts by regulating HIF-1α expression. An amount of 5% O_2_ up-regulates HIF-1α; 1% O_2_ downregulates HIF-1α. In the process of HIF-1α affecting BMSCs osteogenic differentiation, core binding factor α1 (Cbfα1), bone morphogenetic protein 2 (BMP2), runt-related transcription factor 2 (RUNX2), mechano growth factor (MGF), phosphoinositide 3-kinase (PI3K)/protein kinase B (Akt), MEK-extracellular regulated protein kinase 1/2 (ERK1/2), pyruvate dehydrogenase kinase 1 (PDK-1), and Wnt/β-catenin play a positive regulatory role, while Notch1, C/EBP, TWIST, and AKT/mTOR play a negative regulatory role.

**Figure 3 cells-11-03552-f003:**
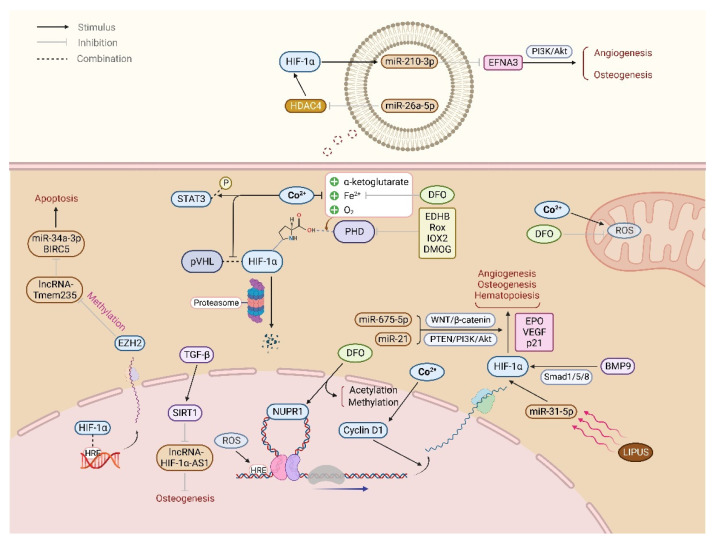
The intracellular regulation mechanism of HIF-1α. Cobalt (Co) ion, deferoxamine (DFO), and other chemicals affect HIF-1α gene expression and hinder HIF-1α degradation by inhibiting the activity of proline hydroxylase (PHD). MiRNAs and lncRNAs participate in the regulation of HIF-1α through a variety of mechanisms, such as Wnt/β-catenin, phosphate, and tension homology deleted on chromosome ten (PTEN)/PI3K/Akt, Smad1/5/8 signal pathways, extracellular vesicles, and epigenetic mechanisms, and exert biological functions such as osteogenesis, angiogenesis, and hematopoiesis.

**Figure 5 cells-11-03552-f005:**
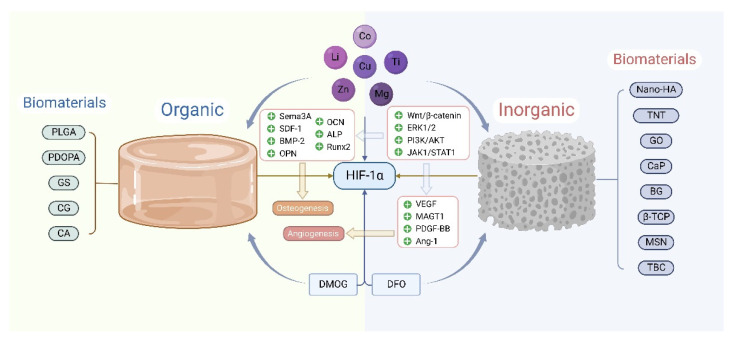
Application of HIF-1α in biomaterials. Organic/inorganic biomaterials can stabilize HIF-1α by doping trace elements such as copper (Cu) and Co or carrying chemical drugs such as dimethyloxalylglycine (DMOG) and DFO. The signal pathways involved include Wnt/β-catenin, ERK1/2, PI3K/AKT, and JAK1/STAT1, which upregulate downstream osteogenic and angiogenic genes.

## Data Availability

Not applicable.

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
