# Peer review of "HIF-1α Regulates Bone Homeostasis and Angiogenesis, Participating in the Occurrence of Bone Metabolic Diseases"

_cells, 2022, doi:10.3390/cells11223552_

Round 1

Reviewer 1 Report

Chen and colleagues' manuscript addresses an interesting topic in the field of metabolic bone disease. However, the  review simply looks like a list of studies without understanding the authors' point of view and may seem unappealing. I would therefore suggest rewriting some parts to make reading more enjoyable.

Also rewrite some unclear sentences, in particular:

lines 59-66

lines 80-88

line 261: Change "BMP9 exists" in "BMP9" is expressed

section 6.1: Please report the target genes of the described miRNAs to better understand the molecular mechanisms involved.

Reviewer 2 Report

Overall the review article is thorough however improvements can be made to strengthen the writing (see below). The authors succeed at providing a comprehensive review of HIFa affects on bone coupling which will be useful for the bone field.  However, The figures need the most improvement. As they are difficult to visualize in many ways (see below) comments.  

Improvements

The writing is not succinct, some sentences are verbose, run-ons, as a result the writing could be made more concise and clear.  Some examples below

114-117, 121-125, 129-133, 167-171, 180-184, 227-232, 245-250, 336-341, 348-351, 377-382, 399-405, 454-459, 459-463, 578-582, 597-601,

All figures needs improvements, the font is hard to read, differences between lines are difficult to denote, symbols &  colors are hard to differentiate

Figure 1- it is not clear what factors contribute to chondrocyte or Adipocyte differentiation as portrayed.

Figure 4- images related to ostoporosis, osteonecrosis, and bone defect appear to have no purpose - at least as imaged.  This figure overall does not summarize what it sets out to do & should be heavily edited to provide better understanding.

Figure 5: please provide a heading above the biomaterials; it is on clear what relationship the trace elements have on the different biomaterials? Is the author stating they are all stimulatory? If so, this is not accurate. At the moment, it is unclear the purpose of the trace elements in this image.

Periods after citations should be removed; gases should have subscripts for the digits: for instance O2 - O2

Round 2

Reviewer 1 Report

The authors addressed all the questions/suggestions and the manuscript has been improved.

Reviewer 2 Report

I have no additional edits, the paper has been heavily edited. 

It is much better in present form.